

# Detecting anomalous electricity consumption with transformer and synthesized anomalies

Tianshi Mu[1], Yun Yu[2], Guocong Feng[2], Huan Luo[1] and Hang Yang[2]

[1] China Southern Power Grid Digital Grid Group Co., Ltd., Guangzhou, China
[2] China Southern Power Grid Co., Ltd., Guangzhou, China

## ABSTRACT

Non-technical losses are consistently a troubling issue for power suppliers. With the application and popularization of smart grid and advanced measurement systems, it has become possible to use data-driven methods to detect anomalous electricity consumption to reduce non-technical losses. A range of machine learning models have been utilized for detecting anomalous electricity consumption and have achieved promising results. However, with the evolution of techniques like electricity theft, coupled with the exponential increase in electricity consumption data, new challenges are constantly being posed for anomalous electricity consumption detection. We propose a Transformer-based method for detecting anomalous electricity consumption. The Transformer is composed of multi-head attention, layer normalization, point-wise feed-forward network, *etc.*, which can effectively handle electricity consumption time-series data. Meanwhile, to alleviate the problem of imbalanced training data between anomalous and normal electricity consumption, we propose a method for synthesizing anomalies. The experimental results demonstrate that our proposed Transformer-based method outperforms the state-of-the-art methods in detecting anomalous electricity consumption, achieving a precision of 93.9%, a recall of 96.3%, an F1-score of 0.951, and an accuracy of 95.6% on a dataset released by the State Grid Corporation of China.

## INTRODUCTION

Electricity has become an indispensable source of energy in our modern life. The normal transmission and usage of electricity is the ultimate goal pursued by power suppliers. However, this is difficult to achieve in reality, as losses frequently occur in the generation, transmission, and distribution of electricity. The losses in electricity can generally be categorized into technical losses and non-technical losses (*De Souza Savian et al., 2021*). Technical losses (*Roselli et al., 2022*) occur during the transmission and distribution of electrical energy through power lines, transformers, and other equipment. These losses are due to factors such as resistance in the wires, transformer inefficiencies, and other technical issues. Technical losses can be minimized by improving the quality of equipment and power supply systems. Non-technical losses (*Messinis & Hatziargyriou, 2018*), on the other hand, refer to electricity being lost due to non-technical reasons such as faulty electrical appliances,

Corresponding author
Tianshi Mu, muts_csg@yeah.net

energy leaks, unauthorized use, and electricity theft. These losses have a significant impact on the overall financial performance of electricity providers. According to a study by the Northeast Group released in October 2021, the electricity theft and non-technical losses cost utilities $101.2 billion annually across 138 countries (*Northeast Group, 2021*). Unlike technical losses, non-technical losses often involve intentional human behavior, making them more difficult to control and prevent.

Fortunately, the smart grid (*Butt, Zulqarnain & Butt, 2021*) is being applied and popularized. The smart grid is an advanced electricity distribution network that utilizes digital communication technology to improve efficiency, reliability, and sustainability. It allows for two-way communication between utilities and consumers (*Faheem et al., 2018*). Advanced metering infrastructure (AMI) (*Fang, Xiao & Wang, 2023*) is a key component of the smart grid. It involves the installation of smart meters that can collect and transmit data on energy consumption at regular intervals, making it possible for real-time monitoring and control of electricity usage. Under such circumstances, data-driven anomaly detection for electricity consumption (*Sun & Zhang, 2020*) has attracted considerable attention. Researchers use big data and machine learning techniques to detect anomalous electricity consumption patterns. This enables proactive steps to address issues such as electricity theft before they escalate into bigger problems, improving energy efficiency practices and reducing non-technical costs.

At present, a range of machine learning models, such as support vector machines (SVM) (*Nagi et al., 2009*; *Depuru, Wang & Devabhaktuni, 2011*; *Nagi et al., 2011*), convolutional neural networks (CNN) (*Zheng et al., 2017*; *Li et al., 2019*) and long short-term memory networks (LSTM) (*Munawar et al., 2021*; *Almazroi & Ayub, 2021*), are being used to detect anomalies in electricity consumption, and these models have achieved promising results. However, with the development of machine learning technology, especially the recent successful application of Transformer (*Vaswani et al., 2017*; *Lin et al., 2022*) in various domains (*Yan et al., 2022*; *Gao et al., 2023*; *Shamshad et al., 2023*), including time series anomaly detection (*Braei & Wagner, 2020*; *Lai et al., 2021*; *Xu et al., 2021*), we realize that the accuracy and efficiency of anomaly detection for electricity consumption can be further improved. In this article, we propose a Transformer-based anomaly detection method for electricity consumption. The Transformer is composed of multiple modules, including multi-head attention, layer normalization, point-wise feed-forward network, and multi-layer perceptron. Meanwhile, to alleviate the problem of imbalanced training data between anomalous and normal electricity consumption, we propose a method for synthesizing anomalies. The experimental results demonstrate that our proposed Transformer-based method outperforms the state-of-the-art methods in detecting anomalous electricity consumption. In summary, we make the following contributions.

- We propose a method for detecting anomalous electricity consumption based on the Transformer.
- We propose a method for synthesizing anomalies to address the problem that normal electricity consumption data is much larger than abnormal data in reality.

- We conduct extensive experiments to evaluate our proposed detection method. The results show that the Transformer-based method outperforms the state-of-the-art methods in detecting anomalous electricity consumption.

The rest of the article is organized as follows. 'Related Work' introduces some related work about the anomaly detection in electricity consumption. 'Method' provides a detailed description of the Transformer-based method for detecting anomalous electricity consumption, including the architecture of the Transformer and the method for synthesizing anomalies. 'Experiments' presents the experimental results. Finally, 'Conclusion' concludes this article.

## RELATED WORK

Electricity consumption anomaly detection has been an important topic of research in recent years. Various approaches have been proposed to detect anomalous electricity consumption. These approaches can be broadly divided into three categories: statistics-based methods, traditional machine learning-based methods, and deep learning-based methods.

The statistics-based methods for electricity consumption anomaly detection are based on the principle that normal electricity usage follows a certain pattern or distribution, and any deviations from this pattern may indicate anomalies. These methods typically involve computing statistical metrics such as mean, standard deviation, median, and percentiles to identify data points that fall outside of expected ranges. For instance, *Li, Bowers & Schnier (2009)* employed the generalized extreme studentized deviate (GESD) and the canonical variate analysis (CVA) to describe latent variables of daily electricity-consumption profiles and to detect the abnormal energy usage. *Badrinath Krishna, Iyer & Sanders (2016)* proposed using the autoregressive moving average (ARIMA) to validate abnormal electricity consumption where smart meter readings are manipulated to steal electricity. The main disadvantage of the statistics-based methods is that they cannot detect complex abnormal electricity consumption patterns.

The traditional machine learning-based methods mainly utilizes traditional machine learning models, including support vector machine (SVM) (*Nagi et al., 2009*; *Depuru, Wang & Devabhaktuni, 2011*; *Nagi et al., 2011*), decision trees (*Cody, Ford & Siraj, 2015*; *Jindal et al., 2016*), K-nearest neighbors (KNN) (*Cai et al., 2017*; *Aziz et al., 2020*), and principal component analysis (PCA) (*Singh, Bose & Joshi, 2017*), to learn the differences between abnormal electricity consumption patterns and normal ones. Although the traditional machine learning-based methods have improved the performance in detecting abnormal electricity consumption compared to statistics-based methods, they require complex feature engineering and the detection accuracy still needs further improvement.

With the development of deep learning technology, deep learning models have also been widely used in anomaly detection of electricity consumption. *Zheng et al. (2017)* proposed a wide and deep convolutional neural network (Wide & Deep CNN) for electricity-theft detection. The Wide & Deep CNN is composed of two components: the wide component and the deep CNN component, enabling it to capture features

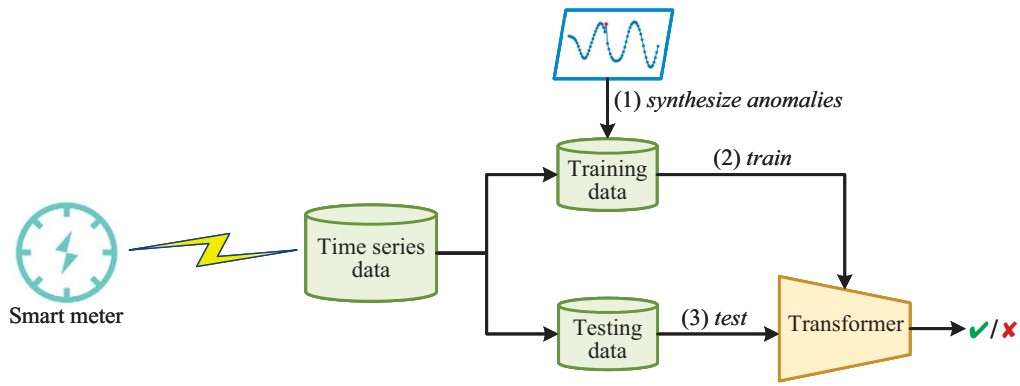

**Figure 1** The framework of detecting anomalous electricity consumption with transformer and synthesized anomalies.

of both 1-D and 2-D electricity consumption data. _Li et al. (2019)_ presented a hybrid convolutional neural network-random forest (CNN-RF) model for automatic electricity theft detection. _Munawar et al. (2021)_ used a hybrid bi-directional GRU and bi-directional LSTM model for electricity theft detection. _Almazroi & Ayub (2021)_ applied the CNN-LSTM technique for detecting electricity fraud, which incorporates the long short-term memory (LSTM) within convolutional neural networks. _Takiddin et al. (2022)_ introduced multiple deep autoencoder-based anomaly detectors for electricity theft detection. The deep learning-based methods can automatically learn from the complex and non-linear relationships in large-scale data and detect subtle anomalies. However, there is still a need for more efficient and accurate methods for handling high-dimensional and complex electricity consumption data.

## METHOD

### Framework

The framework of detecting anomalous electricity consumption with Transformer and synthesized anomalies is shown in Fig. 1. Firstly, the time-series data collected from smart meters is divided into training and testing data. Since normal electricity consumption data is much larger than anomalous electricity consumption data in reality, training the Transformer directly on the imbalanced training data will result in many false negatives. Therefore, in order to improve the ability of the Transformer to detect anomalies, we enhance the training data by synthesizing anomalies. The Transformer used here is a variant of the vanilla Transformer (_Vaswani et al., 2017_). The Transformer incorporates several innovative mechanisms, such as multi-head attention, which enable it to selectively focus on relevant parts of the input sequence. This results in better feature extraction and higher detection rates for various types of abnormal electricity consumption. Once the Transformer is trained, it functions like a simple binary classifier, capable of distinguishing whether the input electricity consumption time-series data is normal or abnormal.

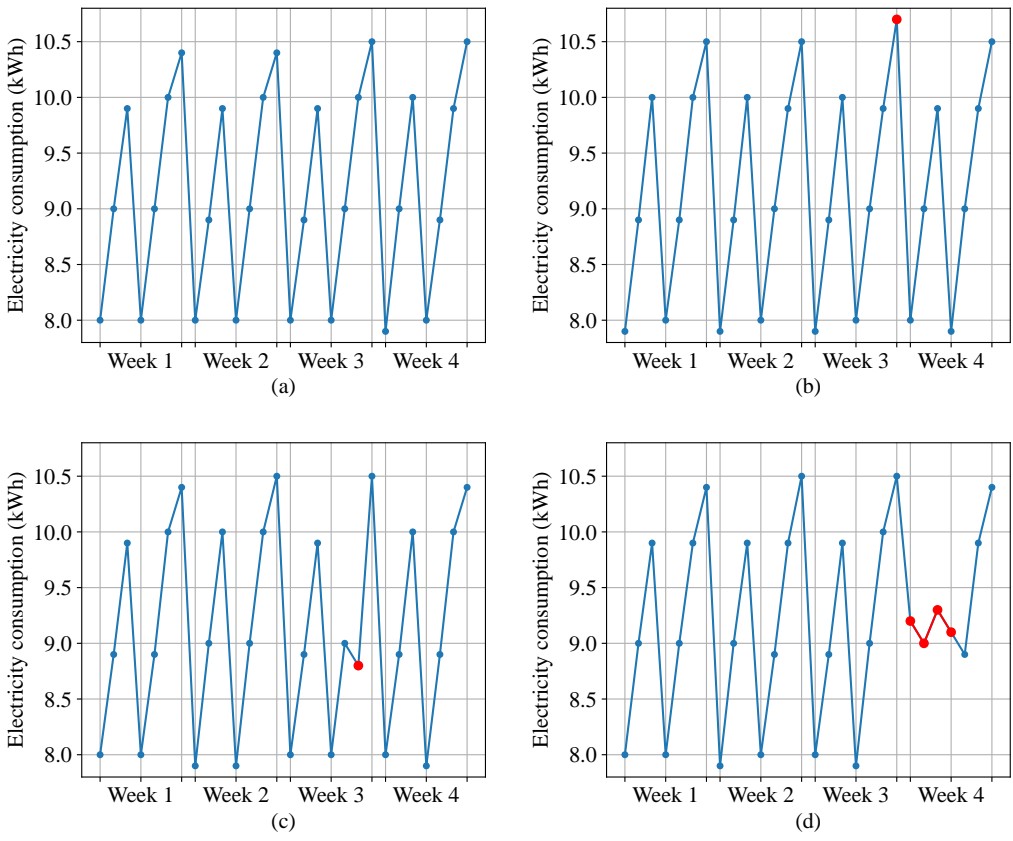

**Figure 2** **Normal electricity consumption and three types of anomalous electricity consumption.** (A) Normal electricity consumption. (B) Point anomalies. (C) Contextual anomalies. (D) Collective anomalies.

## Synthesizing anomalies

Following previous research (*Cook, Mısırlı & Fan, 2019*; *Ruff et al., 2021*; *Zhang, Wu & Boulet, 2021*), we mainly consider three types of anomalies for electricity consumption time-series data, *i.e.,* point anomalies, contextual anomalies, and collective anomalies. We first provide definitions for each type of anomaly, and then describe how to synthesize anomalies for each type.

Figure 2A shows the normal daily electricity consumption of a customer, we can observe that the electricity consumption data fluctuates every day, but exhibits periodicity with a one-week cycle. However, the anomalous electricity consumption takes on the following forms.

**Point anomalies.** These refer to individual data points in a time-series that deviate significantly from the rest of the data. For instance, as shown in Fig. 2B, sudden spikes or dips in electricity consumption can be categorized as point anomalies.

**Contextual anomalies.** These anomalies occur when a specific data point is not anomalous by itself but becomes an anomaly when contextual information is taken into account. For example, as shown in Fig. 2C, based on the electricity data from the

same historical period and the adjacent two days, the third week's sixth day for this customer should show an increasing trend in electricity consumption. However, it displays a decreasing trend instead. Although this data point is not anomalous when compared to other data points, it should be considered as an anomaly when taking into account the contextual information.

**Collective anomalies.** These are characterized by a group of data points that deviate significantly from the normal pattern, but no individual data point stands out as being anomalous on its own. For example, as shown in Fig. 2D, if a high-power electrical appliance malfunctions, the customer may experience a sudden and significant decrease in electricity consumption over a short period of time, and the electricity consumption data during this period should be considered as collective anomalies.

Drawing from the anomaly types defined in the electricity consumption time-series data, we propose a method for synthesizing anomalies. Assuming a time series is $X = (x_1, x_2, \ldots, x_t)$, we can synthesize a point anomaly $x_i$ as follows:

$$x_i = \max(X) + \lambda\sigma(X) \text{ or } x_i = \min(X) - \lambda\sigma(X) \tag{1}$$

where $\sigma(X)$ is the standard deviation of $X$; $\lambda \in (0, 1)$ is a weight parameter that controls how much $x_i$ deviates from the expected value.

Similarly, we can synthesize a contextual anomaly $x_i$ as follows:

$$x_i = \mu(X_{i-k:i+k}) \pm \lambda\sigma(X_{i-k:i+k}) \tag{2}$$

where $\lambda \in (0, 1)$, $X_{i-k:i+k}$ is a subsequence from the time series $X$, and $\mu(X_{i-k:i+k})$ is the mean and $\sigma(X_{i-k:i+k})$ is the standard deviation of $X_{i-k:i+k}$.

The collective anomalies $X_{i:j}$ can be synthesized as follows:

$$X_{i:j} = \lambda X_{i:j} \pm (1 - \lambda)Y \tag{3}$$

where $\lambda \in (0, 1)$, and $Y$ is another time series with the same length as $X_{i:j}$ but a different distribution.

## Transformer-based anomaly detection

As shown in Fig. 3, the Transformer used for detecting anomalous electricity consumption is composed of a stack of $N$ identical Transformer blocks and a multi-layer perceptron. Each Transformer block mainly consists of a multi-head attention module and a point-wise feed-forward network (FFN).

### Multi-head attention

Multi-head attention is a key component of the Transformer. The attention mechanism (*Zhu et al., 2019*) allows the model to selectively focus on parts of the input sequence that are relevant to the current output. In multi-head attention, multiple attention mechanisms operate in parallel with different sets of learned parameters, allowing the model to attend to different aspects of the input.

The attention mechanism in the Transformer is based on the Query-Key-Value model. Given a set of queries $Q$, keys $K$, and values $V$, the multi-head attention operation can be

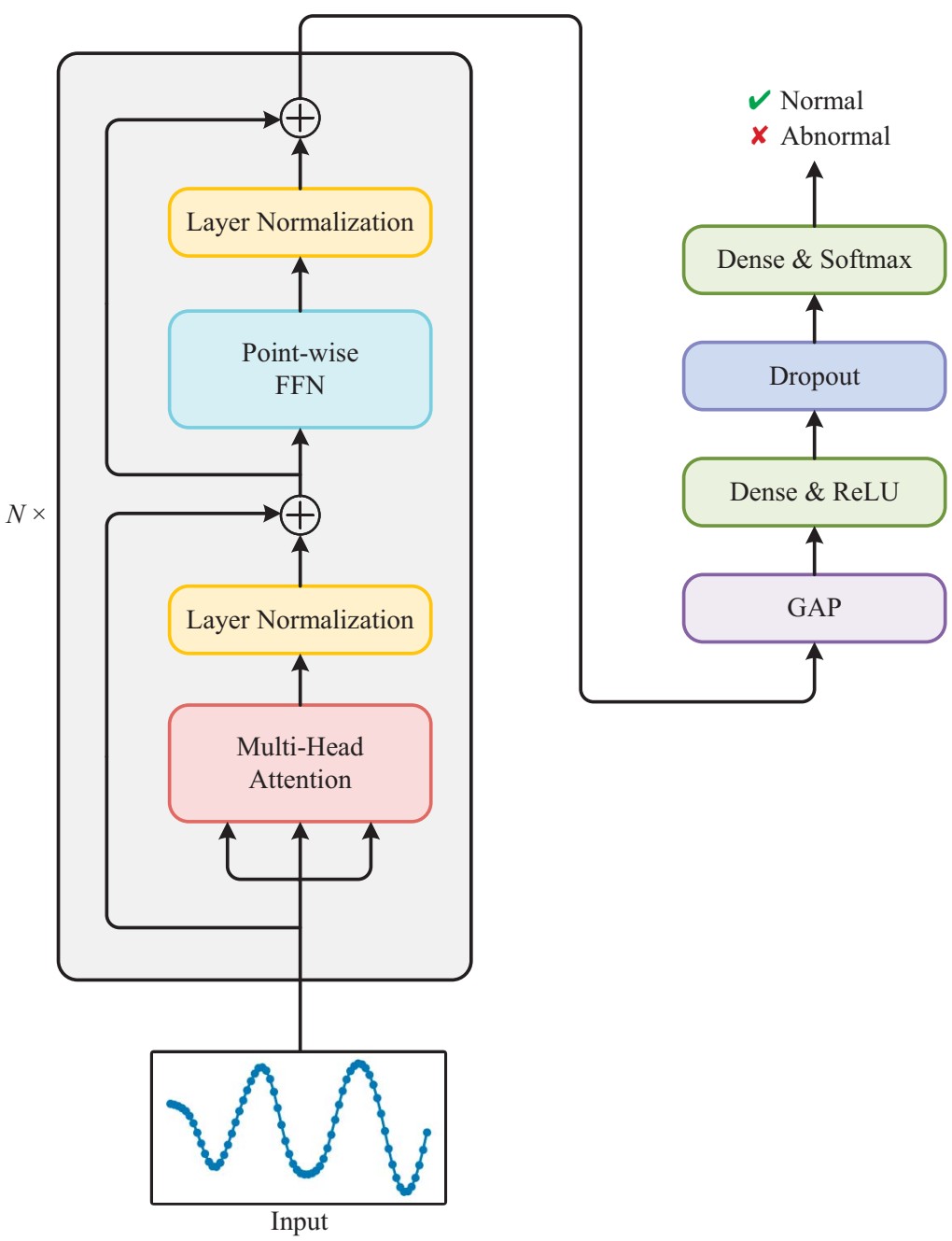

**Figure 3** **Architecture of the Transformer used for detecting anomalous electricity consumption.**

formulated as:

$$head_i = \texttt{Attention}(QW_i^Q, KW_i^K, VW_i^V) \tag{4}$$

where the subscript $i$ indicates the $i$th attention head; $W_i^Q$, $W_i^K$, and $W_i^V$ are learned weight matrices that project the input queries, keys, and values into a common high-dimensional space.

The $\texttt{Attention}$ function computes the weights for each value based on the similarity between its corresponding key and the query vector. The output of each attention head is a weighted sum of the values:

$$\texttt{Attention}(QW_i^Q, KW_i^K, VW_i^V) = \texttt{Softmax}\left(\frac{(QW_i^Q)(KW_i^K)^\top}{\sqrt{d_k}}\right)VW_i^V \tag{5}$$

where $\texttt{Softmax}$ is the Softmax function, $\top$ denotes the transpose operation, and $d_k$ is the dimensionality of the key vectors. The division by $\sqrt{d_k}$ serves as a scaling factor that stabilizes the gradients during training.

The final output of the multi-head attention is obtained by concatenating the outputs of all the individual heads and multiplying them by another learned weight matrix $W^O$:

$$\texttt{MultiHead}(Q, K, V) = \texttt{Concat}(head_1, ..., head_h)W^O \tag{6}$$

where $h$ is the number of attention heads and $\texttt{Concat}$ is the concatenation function. We adopt the self-attention and set $Q = K = V = X$, where $X$ is the outputs of the previous layer.

### Layer normalization

Layer normalization (*Ba, Kiros & Hinton, 2016*) is a technique used in the Transformer to normalize the outputs of each layer before passing them to the next layer. This helps to improve the stability and speed of training by reducing the internal covariate shift, which is the change in the distribution of inputs to a layer that can occur during training.

The layer normalization operation can be formulated as follows:

$$\texttt{LayerNorm}(X) = a\frac{X - \mu}{\sqrt{\sigma^2 + \epsilon}} + b \tag{7}$$

here, $X$ is the input tensor; $\mu$ and $\sigma^2$ are the mean and variance calculated over the last dimension of the input tensor; $\epsilon$ is a small value added to the variance to avoid division by zero; $a$ and $b$ are learned scale and shift parameters, respectively, which are also applied over the last dimension of the input tensor.

### Point-wise FFN

The point-wise feed-forward network (FFN) (*Vaswani et al., 2017*) is a simple yet effective component of the Transformer. It is applied to each position in the sequence independently and identically, which means that it operates on the feature dimension of the input tensor.

The point-wise FFN operation can be formulated as follows:

$$\texttt{FFN}(X) = \texttt{ReLU}(XW_1 + b_1)W_2 + b_2 \tag{8}$$

**Table 1  Experimental data distribution of the SGCC dataset.**

|  |  | Normal (Real) | Abnormal (Real+Synthesized) | Total |
|---|---|---|---|---|
| SGCC | Training | 31,005 | 14,892 (2,892+12,000) | 45,897 |
|  | Testing | 7,752 | 723 (723+0) | 8,475 |
|  | Total | 38,757 | 15,615 (3,615+12,000) | 54,372 |

where $X$ is the input tensor; $W_1$ and $W_2$ are learned weight matrices; $b_1$ and $b_2$ are learned biases; ReLU is the ReLU function.

The point-wise FFN operation consists of two linear transformations. The first transformation projects the input tensor into a higher-dimensional space, while the second transformation maps it back to the original dimensionality. By applying the point-wise FFN operation to each position in the input sequence independently and identically, the model is able to capture different patterns and relationships within the sequence.

### Multi-layer perceptron

The multi-layer perceptron is composed of four layers: Global Average Pooling (GAP), Dense with ReLU activation, Dropout, and Dense with Softmax activation. The GAP layer reduces the spatial dimensions of the input tensor, resulting in a single feature vector. The Dense layer applies linear transformation to this feature vector followed by the ReLU activation function. Then, the Dropout layer randomly drops out some of the neurons, which helps prevent overfitting. Finally, the last Dense layer applies another linear transformation followed by the Softmax activation function, producing the output probabilities for determining whether the input sequence is normal or abnormal.

## EXPERIMENTS

### Experimental settings

We conducted experiments on two electricity consumption datasets, namely the SGCC (State Grid Corporation of China; *Zheng et al., 2017*) dataset and LEAD1.0 (*Gulati & Arjunan, 2022*) dataset. The SGCC dataset, released by the State Grid Corporation of China, is a realistic electricity consumption dataset. It includes the electricity consumption data of 42,372 customers within 1,035 days (from Jan. 1, 2014 to Oct. 31, 2016). It is worth noting that the SGCC dataset contains many missing values, and we employed the method proposed by *Zheng et al. (2017)* to handle these missing values. Among the 42,372 electricity consumption records, there are 38,757 normal records and 3,615 abnormal records. We divided these records into training and testing data in a ratio of 80% and 20%. Since the proportion of abnormal records in the training dataset is very small, we artificially synthesized 12,000 additional abnormal records to enhance the training data. For the SGCC dataset, we only considered the data from the first 365 days; thus the length of the time series is 365. The experimental data distribution of the SGCC dataset is presented in Table 1.

LEAD1.0 dataset is a commercial building hourly electricity consumption dataset, which contains 12,060,910 data points collected by 1,413 smart meters over one year.

**Table 2 Experimental data distribution of the LEAD1.0 dataset.**

|  |  | Normal (Real) | Abnormal (Real+Synthesized) | Total |
|---|---|---|---|---|
|  | Training | 75,912 | 71,914 (47,914+24,000) | 147,826 |
| LEAD1.0 | Testing | 18,978 | 11,978 (11,978+0) | 30,956 |
|  | Total | 94,890 | 83,892 (59,892+24,000) | 178,782 |

**Table 3 The key parameter settings for different methods.**

| Method | Parameters |
|---|---|
| SVM (*Depuru, Wang & Devabhaktuni, 2011*) | Kernel type: RBF kernel; regularization parameter $C = 1$; Kernel coefficient $\gamma = 0.92$. |
| Wide & Deep CNN (*Zheng et al., 2017*) | The size of the wide component $\alpha = 60$; the size of the CNN component $\beta = 120$; the number of filters $\gamma = 15$. |
| LSTM (*Munawar et al., 2021*) | Embedding dimension $d_e = 256$; the number of hidden units $n_h = 128$. |
| CNN-LSTM (*Almazroi & Ayub, 2021*) | Embedding dimension $d_e = 256$; the number of hidden units $n_h = 128$. |
| Autoencoder (*Takiddin et al., 2022*) | Encoder Layers $n_e = 4$; decoder layers $n_d = 2$. |
| Transformer (Ours) | Key dimension $d_k = 256$; number of heads $h = 4$. |

For the LEAD1.0 dataset, we have chosen the 360-hour electricity consumption data from each commercial building as the time-series data, resulting in a time-series length of 360. We finally selected 94,890 normal electricity consumption time-series data and 59,892 abnormal electricity consumption time-series data from the LEAD1.0 dataset as the experimental data. Additionally, we also synthesized 24,000 anomalies to enhance the training data. The experimental data distribution of the LEAD1.0 dataset is presented in Table 2.

Due to the nature of the experimental datasets, some sequences are only labeled as anomalous or not, without specifying the specific points that are anomalous. Therefore, our detection method focuses on determining whether the entire sequence is anomalous, rather than identifying the specific types of anomalies (*e.g.*, point anomalies, contextual anomalies, and collective anomalies) occurring within it.

In order to provide a comprehensive evaluation of the detection method we proposed, we compared it with five state-of-the-art methods, *i.e.,* SVM (*Depuru, Wang & Devabhaktuni, 2011*), Wide & Deep CNN (*Zheng et al., 2017*), LSTM (*Munawar et al., 2021*), CNN-LSTM (*Almazroi & Ayub, 2021*), and Autoencoder (*Takiddin et al., 2022*). The key parameter settings for these methods are shown in Table 3.

### Evaluation metrics

We adopted five commonly used evaluation metrics: ROC curve, precision, recall, F1-score, and accuracy. As we are detecting anomalous electricity consumption, in this article, "positive" refers to anomalous consumption records, while "negative" refers to normal consumption records.

ROC curve is a graphical representation of the true positive rate (*TPR*) against the false positive rate (*FPR*) at different classification thresholds. It helps in choosing an optimal threshold for a model, considering the trade-off between true positives and false positives.

Precision is the fraction of true positives out of all the predicted positives. It tells us how often a model correctly predicts positive cases. A high precision score means that the model has fewer false positives. Recall is the fraction of true positives out of all the actual positives. It tells us how well a model can identify positive cases. A high recall score means that the model has fewer false negatives. F1-score is the harmonic mean of precision and recall. It provides a balance between precision and recall. Accuracy is used to measure the proportion of correct predictions made by a model. It provides an overall assessment of the model's performance in terms of correctly classifying instances.

If we use *TP* to represent the number of true positives, *FP* to represent the number of false positives, *TN* to represent the number of true negatives, and *FN* to represent the number of false negatives, then the formulas for the above metrics are as follows.

$$Precision = \frac{TP}{TP + FP} \tag{9}$$

$$Recall\ (TPR) = \frac{TP}{TP + FN} \tag{10}$$

$$F1\text{-}score = 2 * \frac{Precision * Recall}{Precision + Recall} \tag{11}$$

$$FPR = \frac{FP}{FP + TN} \tag{12}$$

$$Accuracy = \frac{TP + TN}{TP + FP + TN + FN} \tag{13}$$

## Experimental results

The ROC curves of the six methods are shown in Fig. 4. As we can observe, the overall trend of the ROC curves is consistent on both the SGCC dataset and the LEAD1.0 dataset. The ROC curves demonstrate that our proposed Transformer has the largest area under the curve (AUC) among all the methods, indicating its superiority in detecting anomalous electricity consumption. The SVM has the poorest detection performance. Moreover, it can be observed that for each method, the *TPR* increases as the *FPR* increases. To ensure a fair comparison in subsequent experiments, we will record the performance metrics of each method at a *FPR* of 0.5%.

The performance comparison of different methods on the SGCC dataset is shown in Table 4. The precision of different methods is relatively high, all above 90%, indicating that all methods can accurately detect abnormal electricity consumption. However, in terms

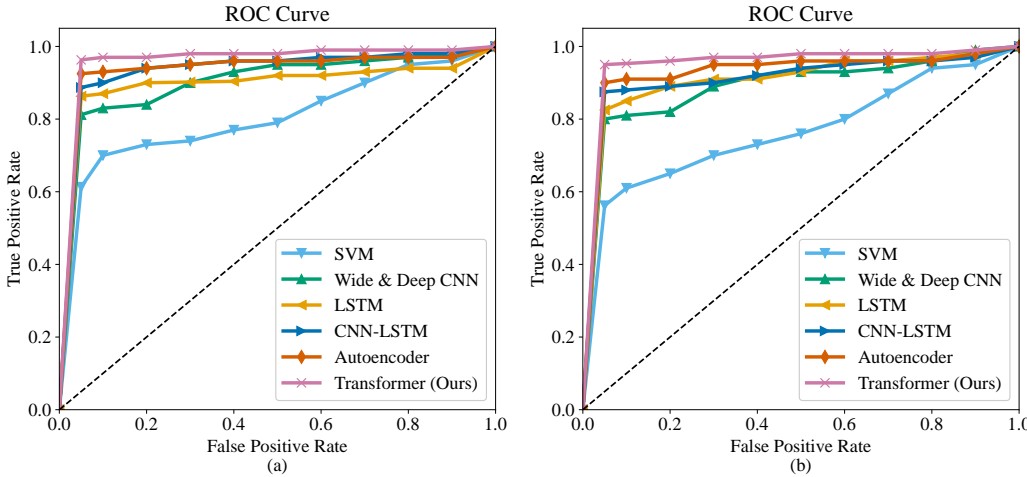

**Figure 4** **ROC curves of different methods.** (A) ROC curve on the SGCC dataset. (B) ROC curve on the LEAD1.0 dataset.

**Table 4** **Performance comparison of different methods on the SGCC dataset ( *FPR* = 0.5%).**

| Dataset | Model | Precision | Recall | F1-score | Accuracy |
|---------|-------|-----------|--------|----------|----------|
| SGCC | SVM (*Depuru, Wang & Devabhaktuni, 2011*) | 0.907 | 0.613 | 0.731 | 0.800 |
| | Wide & Deep CNN (*Zheng et al., 2017*) | 0.929 | 0.812 | 0.867 | 0.889 |
| | LSTM (*Munawar et al., 2021*) | 0.932 | 0.863 | 0.896 | 0.911 |
| | CNN-LSTM (*Almazroi & Ayub, 2021*) | 0.934 | 0.887 | 0.910 | 0.922 |
| | Autoencoder (*Takiddin et al., 2022*) | 0.937 | 0.925 | 0.931 | 0.939 |
| | Transformer (Ours) | 0.939 | 0.963 | 0.951 | 0.956 |

of recall, F1-score and accuracy, there are significant differences in the performance of different methods. Specifically, the Transformer we proposed exhibits the best performance, with a recall of 96.3%, an F1-score of 0.951, and an accuracy of 95.6%. The Autoencoder performs the second best, with a recall of 92.5%, an F1-score of 0.931, and an accuracy of 93.1%. The SVM has the worst performance, with a recall of only 61.3%, an F1-score of 0.731, and an accuracy of 80.0%.

The performance comparison of different methods on the LEAD1.0 dataset is shown in Table 5. We can see that the experimental results on the LEAD1.0 dataset are consistent with those on the SGCC dataset. Our proposed Transformer still performs the best, with a precision of 93.8%, a recall of 95.0%, an F1-score of 0.944, and an accuracy of 95.0%. It should be noted that for the same method, the results on the LEAD1.0 dataset are somewhat inferior to those on the SGCC dataset, suggesting that anomalous electricity consumption patterns in the LEAD1.0 dataset are more challenging to detect than those in the SGCC dataset.

In summary, the experimental results on both the SGCC dataset and the LEAD1.0 dataset demonstrate that our proposed Transformer outperforms the state-of-the-art methods in

**Table 5  Performance comparison of different methods on the LEAD1.0 dataset (*FPR* = 0.5%).**

| Dataset | Model | Precision | Recall | F1-score | Accuracy |
|---------|-------|-----------|--------|----------|----------|
| LEAD1.0 | SVM (*Depuru, Wang & Devabhaktuni, 2011*) | 0.900 | 0.562 | 0.692 | 0.778 |
| | Wide & Deep CNN (*Zheng et al., 2017*) | 0.928 | 0.800 | 0.859 | 0.883 |
| | LSTM (*Munawar et al., 2021*) | 0.930 | 0.825 | 0.874 | 0.894 |
| | CNN-LSTM (*Almazroi & Ayub, 2021*) | 0.933 | 0.875 | 0.903 | 0.917 |
| | Autoencoder (*Takiddin et al., 2022*) | 0.935 | 0.900 | 0.917 | 0.928 |
| | Transformer (Ours) | 0.938 | 0.950 | 0.944 | 0.950 |

detecting anomalous electricity consumption, achieving higher precision, recall, F1-score, and accuracy.

## Ablation study

In our proposed method for detecting abnormal electricity consumption based on the Transformer, we artificially synthesized anomalies to enhance the training dataset to alleviate the imbalance between normal and abnormal electricity consumption patterns. To verify whether the synthesized anomalies can improve the performance of the Transformer, we conducted an ablation study in this section. The experimental results are shown in Fig. 5. It can be observed that training the Transformer with synthesized anomalies does not have a significant impact on its precision on both the SGCC and LEAD1.0 datasets, but significantly improves its recall and F1-score. For example, on the SGCC dataset, when training the Transformer with additional synthesized anomalies, the precision increases slightly from 92.6% to 93.9%, while the recall increases from 78.7% to 96.3%, F1-score increases from 0.851 to 0.951, and the accuracy increases from 87.8% to 95.6%. We analyzed that the reason for this observation might be that, without the additional synthesized anomalies, the training dataset has a significantly higher number of normal electricity consumption patterns than abnormal electricity consumption patterns. Consequently, training the Transformer on this imbalanced dataset causes it to overfit heavily to normal electricity consumption patterns. As a result, it produces fewer false alarms but misses many abnormal electricity consumption patterns during detection, leading to a high precision but a low recall. Therefore, synthesizing anomalies is very helpful in improving the performance of the Transformer.

## CONCLUSION

In this article, we propose a Transformer-based method for detecting anomalous electricity. In particular, the Transformer is composed of multiple modules, including multi-head attention, layer normalization, point-wise feed-forward network, and multi-layer perceptron. Since normal electricity consumption data is much larger than abnormal consumption data in reality, we propose a method for artificially synthesizing anomalies to alleviate the issue of imbalanced training data. To comprehensively evaluate the method we proposed, we conduct experiments on two representative datasets and also compare it with the state-of-the-art methods. The experimental results demonstrate that our proposed method outperforms other methods and can effectively detect anomalies in electricity

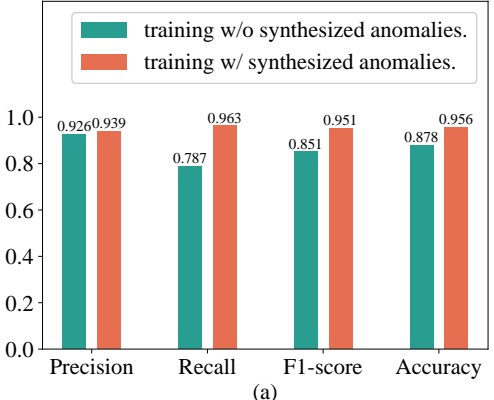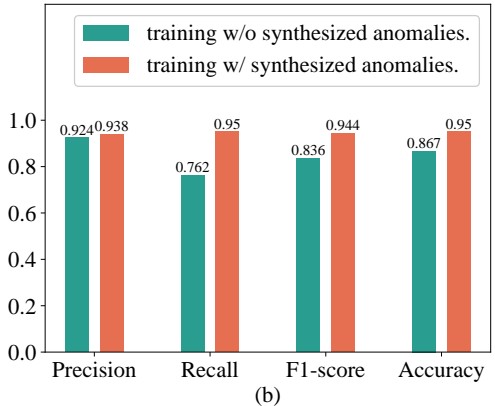

**Figure 5    Performance of the Transformer trained with and without synthesized anomalies.** (A) Results on the SGCC dataset. (B) Results on the the LEAD1.0 dataset.

consumption. In the future, we will further explore how to leverage Transformers to detect various anomalies in electricity consumption more precisely, improve their performance on imbalanced training data, and employ them for forecasting the anomalous electricity consumption.

### Funding
This work was supported by the China Southern Power Grid Co., Ltd. The funders had no role in study design, data collection and analysis, decision to publish, or preparation of the manuscript.

### Grant Disclosures
The following grant information was disclosed by the authors:
The China Southern Power Grid Co., Ltd.

### Competing Interests
Tianshi Mu and Huan Luo are employed by China Southern Power Grid Digital Grid Group Co., Ltd and Yun Yu, Guocong Feng and Hang Yang are employed by China Southern Power Grid Co., Ltd.

### Author Contributions
- Tianshi Mu conceived and designed the experiments, performed the computation work, authored or reviewed drafts of the article, and approved the final draft.
- Yun Yu performed the experiments, analyzed the data, prepared figures and/or tables, and approved the final draft.
- Guocong Feng conceived and designed the experiments, authored or reviewed drafts of the article, and approved the final draft.

- Huan Luo performed the experiments, performed the computation work, prepared figures and/or tables, and approved the final draft.
- Hang Yang analyzed the data, prepared figures and/or tables, and approved the final draft.

## Data Availability

The data is available at GitHub:

- https://github.com/henryRDlab/ElectricityTheftDetection.
- https://github.com/samy101/lead-dataset.

## Supplemental Information

Supplemental information for this article can be found online at http://dx.doi.org/10.7717/peerj-cs.1721#supplemental-information.

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
