# Peer review of "Detecting anomalous electricity consumption with transformer and synthesized anomalies"

_PeerJ Computer Science, doi:10.7717/peerj-cs.1721_

## Round 0.1 · original submission · Major Revisions

This manuscript was reviewed by three experts, all of whom raised serious concerns. The authors should put efforts into enhancing the novelty of the proposed work and justify the usage of test data.

Reviewers 2 & 3 have suggested that you cite specific references. You are welcome to add it/them if you believe they are relevant. However, you are not required to include these citations, and if you do not include them, this will not influence my decision.

**Language Note:** The review process has identified that the English language must be improved. PeerJ can provide language editing services - please contact us at copyediting@peerj.com for pricing (be sure to provide your manuscript number and title). Alternatively, you should make your own arrangements to improve the language quality and provide details in your response letter. – PeerJ Staff

Reviewer 1 ·

Basic reporting

The literature is clear and professional English is used throughout. However, there is already much research on anomalous electricity consumption detection based on Transformer. What is the innovation of the proposed method in this paper?

Experimental design

1. What is the purpose of anomaly detection? Is the final output of the model to discern whether a particular time series is anomalous or is it able to identify which specific data points are anomalous?
2. What is the time resolution and the length of the training time series? Please provide more detailed information on the training dataset.

Validity of the findings

no comment

Additional comments

no comment

Reviewer 2 ·

Basic reporting

Relevant prior literature should be appropriately referenced.

Experimental design

Research question well defined, relevant & meaningful.

Validity of the findings

no comment

Additional comments

This paper proposed a Transformer-based method for detecting anomalous electricity consumption. The following problems still exist, and the author is suggested to revise them carefully.
1. The experimental results achieved a precision of 93.9%, a recall of 96.3%, and an F1-score of 0.951 on the SGCC dataset. The result is basically similar to the data results of LEAD1.0 dataset. What is the relationship between these two data?
2. This paper proposed a method for detecting anomalous electricity consumption based on the Transformer, Could the author explain where the specific innovation lies?
3. What are the characteristics of the data of anomalous electricity consumption? Has the author considered extracting its features?
4. This paper compared the results with five state-of-the-art methods, i.e., SVM (Depuru et al., 2011), Wide & Deep CNN (Zheng et al., 2017), LSTM (Munawar et al., 2021), CNN-LSTM (Almazroi and Ayub, 2021), and Autoencoder (Takiddin et al., 2022). May I ask if the author directly compares the results with those in their article, or uses the methods in their papers to detect the data in this paper? What are the parameters of these methods?
5. The difference between the results of this method and some algorithms(such as Autoencoder) is not obvious. Does the method have any other advantages?
6. Regarding transformer, please refer to the latest literature below, and it is recommended to discuss and compare it in the introduction.
[1] Gao, Y., Gong, G., Ye, B., Tian, X., Li, N., & Yuan, H. Leveraging knowledge graph for domain-specific Chinese named entity recognition via lexicon-based relational graph transformer, International Journal of Bio-Inspired Computation, 2023, Vol. 21, No. 3, pp 148-162, https://doi.org/10.1504/IJBIC.2023.131912
[2] Yan, L., Mu, G., Wang, Q., He, Z., & Zhu, Y. Time sequence information-based transformer for the judgement on the state of power dispatching. International Journal of Computing Science and Mathematics. 2022, Vol. 16, No. 1, pp 71–84. https://doi.org/10.1504/IJCSM.2022.126807

Reviewer 3 ·

Basic reporting

* Literature studies are insufficient. Please review more studies. One of the intellectual studies in this field is given below.

"https://doi.org/10.1016/j.compag.2020.105559"

* Correct the errors by reviewing the language of the article.

* Why was Accuracy not used in Evaluation metrics? This metric is generally used when comparing literature studies.

Experimental design

* Despite synthesizing the SGCC dataset, there is still a serious imbalance between the number of anomaly data and the number of normal data. Wouldn't that negatively impact performance metrics?

* No reference is added in Section 2, where the methods are described. Appropriate references should be included in the section between lines 114-210.

Validity of the findings

* Has the Cross-Validation approach been taken into account for the performance calculation after the training and test sets have been created?

Additional comments

* For the code part, an explanation should be made using an extra pdf.

---

## Round 0.2 · accepted · Accept

Three reviewers were involved in the current review process, and they are satisfied with the revisions made by the authors. Although one reviewer suggested two studies, the authors are not required to cite them if they have no direct relation to the proposed work.

Reviewer 1 ·

Basic reporting

no comment

Experimental design

no comment

Validity of the findings

no comment

Additional comments

The comments have been well addressed.

Reviewer 2 ·

Basic reporting

no comment

Experimental design

no comment

Validity of the findings

no comment

Additional comments

no comment

Reviewer 3 ·

Basic reporting

No Comment

Experimental design

No Comment

Validity of the findings

No Comment

Additional comments

Dear Author, First of all, thank you for answering most of the questions. However, you could not fully relate the article I presented to you as a suggestion to your own work. The following two recent studies in the literature emphasize the importance of deep learning and machine learning in locating illegal transformers to prevent electricity theft, especially in agricultural areas. The reason why I recommend these studies is to provide basic information about preventing electricity theft , especially in the literature section. I suggest you review the studies again.

Regards,

1- doi: 10.1109/ACCESS.2023.3323694.
2- https://doi.org/10.1016/j.compag.2020.105559